# $\ell_1$ ADVERSARIAL ROBUSTNESS CERTIFICATES: A RANDOMIZED SMOOTHING APPROACH

## ABSTRACT

Robustness is an important property to guarantee the security of machine learning models. It has recently been demonstrated that strong robustness certificates can be obtained on ensemble classifiers generated by input randomization. However, tight robustness certificates are only known for symmetric norms including $\ell_0$ and $\ell_2$, while for asymmetric norms like $\ell_1$, the existing techniques do not apply. By converting the likelihood ratio into a one dimensional mixed random variable, we derive the first tight $\ell_1$ robustness certificate under isotropic Laplace distributions in binary case. Empirically, the deep networks smoothed by Laplace distributions yield the state-of-the-art certified robustness in $\ell_1$ norm on CIFAR-10 and ImageNet.

## 1 INTRODUCTION

have done a series of nice works in practical sights or theoretical sights (Zheng et al., 2016; Gouk et al., 2018). Among them, certifiably robustness is valuable, since it can withstand all attacks within a norm ball and has a nice theoretical and practical outcome. However, most work cannot deal with the case for general neural networks.

Deep networks are flexible models that are widely adopted in various applications. However, it has been shown that such models are vulnerable against adversary (Szegedy et al., 2014). Concretely, an unnoticeable small perturbation on the input can cause a typical deep model to change predictions arbitrarily. The phenomenon raises the concerns of the security of deep models, and hinders its deployment in decision-critical applications. Indeed, the certification of robustness is a pre-requisite when AI-generated decisions may have important consequences.

Certifying the robustness of a machine learning model is challenging, especially for modern deep learning models that are over-parameterized and effectively black-box. Hence, the existing approaches mainly rely on empirical demonstration against specific *adversarial attack* algorithms (Goodfellow et al., 2015; Madry et al., 2018; Finlay et al., 2019). However, this line of works can give a false sense of security. Indeed, successful defense against the existing attack algorithms does not *guarantee* actual robustness against any adversaries that may appear in the future.

Recently, the adversarial robustness community has shifted the focus towards establishing certificates that prove the robustness of deep learning models. The certificate can be either exact or conservative, so long as the certified region cannot exhibit any adversarial examples. Given the over-parameterized deep models and modern high-dimensional datasets, scalability becomes a key property for the certification algorithms, as many methods are computationally intractable.

Our work is based on the novel modeling scheme that generates ensembles of a fixed black-box classifier based on input randomization (Cohen et al., 2019). Under this framework, tight robustness certificates can be obtained with only the ensemble prediction values and randomization parameters. Given appropriate choices of distributions, the robustness guarantee can be derived for $\ell_2$ or $\ell_0$ norms (Cohen et al., 2019; Lee et al., 2019). The tightness simply implies that any point outside the certified region is an adversarial example in the worst case. However, the derivations of the previous results heavily relies on the fact that the target norm ($\ell_2$ or $\ell_0$) is symmetric, therefore analyzing any perturbation direction for attacking the model gives the same certification guarantee.

In contrast, $\ell_1$ norm is asymmetric. That is, for a given $\ell_1$ ball centered at the origin, if we move another $\ell_1$ ball also from the origin by a distance $\boldsymbol{\delta}$, where $\|\boldsymbol{\delta}\|_1$ is fixed, then the overlapped region

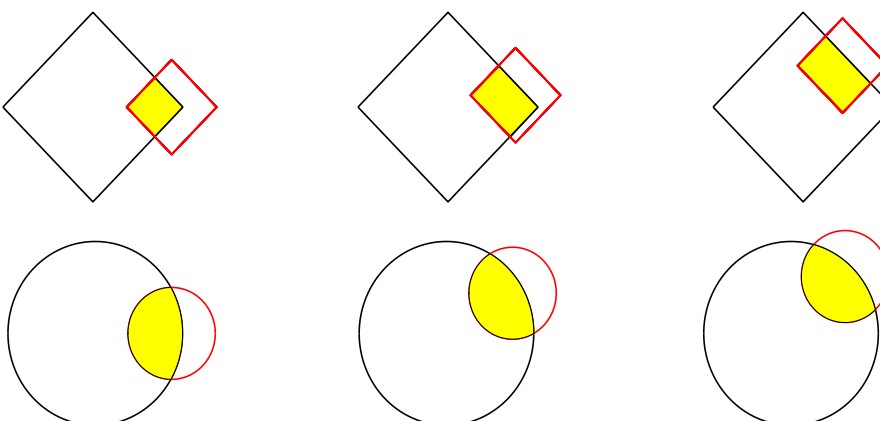

Figure 1: In $\ell_1$ case, if we perturb the input with $\boldsymbol{\delta}$ such that $\|\boldsymbol{\delta}\|_1$ is fixed, we may get very different overlapped regions with different size. Notice that this is different from $\ell_2$ (or $\ell_0$, not shown), where the overlapped regions are always symmetric and of the same size.

between the two $\ell_1$ balls may have different shapes and sizes (See Figure 1). The characterization of this overlapped region is the key step for proving tight certificates, hence the existing techniques do not apply for $\ell_1$ norm.

In this work, we derive a tight $\ell_1$ robustness guarantee under isotropic Laplace distributions. The Laplace distribution can be interpreted as an infinite mixture of uniform distributions over $\ell_1$-norm balls, which is a natural "conjugate" distribution for $\ell_1$ norm. Due to asymmetry, we first identified the tight robustness certificate for attacking the model in one particular direction, $\boldsymbol{\delta} = (\|\boldsymbol{\delta}\|_1, 0, \cdots, 0)$. To show that other perturbation directions cannot lead to worse results, we convert the $d$ dimensional likelihood function into an one dimensional function, where we apply relaxation for various $\boldsymbol{\delta}$ and show that the worst case result is bounded by the specific direction $(\|\boldsymbol{\delta}\|_1, 0, \cdots, 0)$.

Theoretically, our certificate is *tight* in the binary classification setting. In the multi-class classification setting, our certificate is always tighter than the previous certificate proposed by Lecuyer et al. (2019). The theoretical improvement always leads to superior empirical results on certifying the same model, where we demonstrate the result on CIFAR-10 and ImageNet with ResNet models. Moreover, the proposed robustness certificate on models smoothed by Laplace distributions also outperforms the same models trained and certified using Gaussian distributions (Cohen et al., 2019) in $\ell_1$ certified robustness, where the Gaussian-based robustness certificate is adapted from $\ell_2$ norm.

## 2 RELATED WORK

Robustness of a model can be defined in various aspects. For example, Feynman-Kac Formalism can be used to improve robustness (Wang et al., 2018). In this paper, we focus on the classification setting, where the goal is to provide guarantee of a constant prediction among a small region specified via some metric. The robustness certificate can be either exact or conservative, so long as a constant prediction is guaranteed in the certified region. Note that the certification of a completely black-box model requires checking the prediction values at every point around the point of interest, which is clearly infeasible. A practical certification algorithm inevitably has to specify and leverage the functional structure of the classifier in use to reduce the required computation.

**Exact certificates.** The exact certificate of deep networks is typically derived for the networks with a piecewise linear activation function such as ReLU. Such networks have an equivalent mixed integer linear representation (Cheng et al., 2017; Lomuscio & Maganti, 2017; Dutta et al., 2017; Bunel et al., 2018). Hence, one may apply mixed integer linear programming to find the worst case adversary within any convex polyhedron such as an $\ell_1$-ball or $\ell_\infty$-ball. Despite the elegant solution, the complexity is, in general, NP-hard and the algorithms are not scalable to large problems(Tjeng et al., 2017).

**Conservative certificates.** A conservative certificate can be more scalable than the exact methods, since one can trade-off the accuracy of certification with efficiency (Gouk et al., 2018; Tsuzuku et al., 2018; Cisse et al., 2017; Anil et al., 2018; Hein & Andriushchenko, 2017). For example, one can relax the search of the worst case adversary as a simpler optimization problem that only bounds the effect of such adversary. Alternatively, people also consider the robustness problem in a modular way, where the robustness guarantee can be derived iteratively for each layer in the deep networks by considering the feasible values for each hidden layer (Gowal et al., 2018; Weng et al., 2018; Zhang et al., 2018; Mirman et al., 2018; Singh et al., 2018). However, this line of works have not yet been demonstrated to be feasible to realistic networks in high dimensional problems like ImageNet.

**Randomized smoothing.** Randomized smoothing has been proved to be closely related to robustness. Although similar techniques have been tried by (Liu et al., 2018; Cao & Gong, 2017), no corresponding proofs have been given; Li et al. (2018) and Cohen et al. (2019) have proved certified robustness of $\ell_2$ norm under isotropic Gaussian noise, and Lee et al. (2019) proved robustness for $\ell_0$ form. Lecuyer et al. (2019) use techniques from differential privacy to prove $\ell_1$ robustness under Gaussian and Laplace noise respectively, but the bounds are not tight. Li et al. (2018); Pinot et al. (2019) use Rényi divergence framework without tightness proof. Our results synthesize the ideas in (Cohen et al., 2019; Lee et al., 2019; Lecuyer et al., 2019; Li et al., 2018; Pinot et al., 2019) and prove the tight robustness radius under the binary classification setting.

## 3 PRELIMINARIES

**Definition 1 (Laplace distribution)** *Given $\lambda \in \mathbb{R}^+$, $d \in \mathbb{Z}^+$, we use $\mathcal{L}(\lambda)$ to denote the Laplace distribution in dimension $d$ with parameter $\lambda$. The p.d.f. of $\mathcal{L}(\lambda)$ is denoted as $\mathcal{L}(\boldsymbol{x}; \lambda) \triangleq \frac{1}{(2\lambda)^d} \exp(-\frac{\|\boldsymbol{x}\|_1}{\lambda})$.*

As we will see in Lemma 3.1, in smoothing analysis, we are interested in the likelihood ratio of two random variables $X = \boldsymbol{\epsilon}$ and $Y = \boldsymbol{\delta} + \boldsymbol{\epsilon}$ (here $\boldsymbol{\epsilon} \sim \mathcal{L}(\lambda)$ and $\boldsymbol{\delta} \in \mathbb{R}^d$ is a fixed vector). Specifically,

$$\frac{\mu_Y(\boldsymbol{x})}{\mu_X(\boldsymbol{x})} = \exp\left(-\frac{1}{\lambda}(\|\boldsymbol{x} - \boldsymbol{\delta}\|_1 - \|\boldsymbol{x}\|_1)\right)$$

Therefore, the likelihood ratio between two $d$ dimensional random variables is controlled by a one dimensional random variable $T(\boldsymbol{x}) \triangleq \|\boldsymbol{x} - \boldsymbol{\delta}\|_1 - \|\boldsymbol{x}\|_1$, where $\boldsymbol{x} \sim \mathcal{L}(\lambda)$. This transformation is crucial in our analysis, and it is easy to see that $T(\boldsymbol{x})$ is a mixed random variable, since $\mathbb{P}_{\boldsymbol{x}}(T(\boldsymbol{x}) = \|\boldsymbol{\delta}\|_1) > 0$.

In our analysis, we need to calculate the inverse of c.d.f. of $T(x)$. However, since $T(x)$ is a mixed random variable, sometimes the inverse may not exist. See Figure 3 for illustration, where the inverse of the probability $0.85$ does not exist. To deal with this case, we have the following modified version of Neyman-Pearson Lemma, with the proof in Appendix A.

**Lemma 3.1** *(Neyman-Pearson Lemma for mixed random variables). Let $X \sim \mathcal{L}(\lambda)$ and $Y \sim \mathcal{L}(\lambda) + \boldsymbol{\delta}$. Let $h : \mathbb{R}^d \to \{0, 1\}$ be any deterministic or random function. Given any $\beta \in \mathbb{R}$, and $S' \subseteq \{\boldsymbol{z} \in \mathbb{R}^d : \|\boldsymbol{z} - \boldsymbol{\delta}\|_1 - \|\boldsymbol{z}\|_1 = \beta\}$:*

*1. If $S = \{\boldsymbol{z} \in \mathbb{R}^d : \|\boldsymbol{z} - \boldsymbol{\delta}\|_1 - \|\boldsymbol{z}\|_1 > \beta\} \cup S'$, and $\mathbb{P}(h(X) = 1) \geq \mathbb{P}(X \in S)$ then $\mathbb{P}(h(Y) = 1) \geq \mathbb{P}(Y \in S)$*

*2. If $S = \{\boldsymbol{z} \in \mathbb{R}^d : \|\boldsymbol{z} - \boldsymbol{\delta}\|_1 - \|\boldsymbol{z}\|_1 < \beta\} \cup S'$, and $\mathbb{P}(h(X) = 1) \leq \mathbb{P}(X \in S)$, then $\mathbb{P}(h(Y) = 1) \leq \mathbb{P}(Y \in S)$*

## 4 MAIN RESULTS

In this paper, we apply the randomized smoothing technique (Cohen et al., 2019) for getting robustness certificates, which works as follows. Given an input $x$, we perturb it with $\boldsymbol{\epsilon}$, s.t. $\boldsymbol{\epsilon} \sim \mathcal{L}(\lambda)$. Then instead of evaluating the robustness of the original function $f(\boldsymbol{x})$, we evaluate $g(\boldsymbol{x}) \triangleq \arg\max_c \mathbb{P}_{\boldsymbol{\epsilon}}(f(\boldsymbol{x} + \boldsymbol{\epsilon}) = c)$, which is effectively the smoothed version of $f(\boldsymbol{x})$.

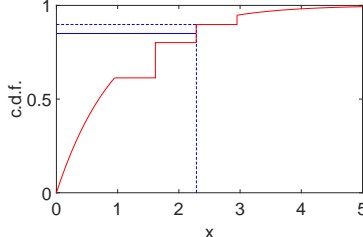 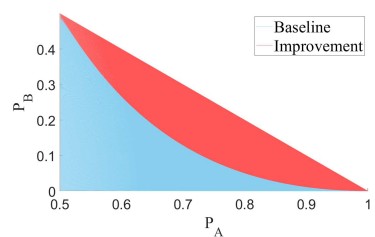

Figure 3: For mixed random variables, sometimes the inverse of the probability does not exist. E.g., see the solid blue line.

Figure 4: Comparison for Eqn. (1). Green region shows that baseline is better, while red region shows our new bound is better.

### 4.1 ROBUSTNESS CERTIFICATES FOR GENERAL CASES

Our first theorem proves that for the smoothed classifier $g$, and a given input $x$, there always exists a robust radius $R$, such that any perturbation $\delta$ s.t. $\|\delta\|_1 \leq R$, does not alter the prediction of $g(x)$.

**Theorem 1** *Let $f : \mathbb{R}^d \to Y$ be deterministic or random function, and let $\epsilon \sim \mathcal{L}(\lambda)$. Let $g(x) = \arg\max_c \mathbb{P}_\epsilon(f(x + \epsilon) = c)$. Suppose $\underline{P_A}, \overline{P_B} \in [0, 1]$ are such that*

$$\mathbb{P}\left(f(x + \epsilon) = c_A\right) \geq \underline{P_A} \geq \overline{P_B} \geq \max_{c \neq c_A} \mathbb{P}(f(x + \epsilon) = c)$$

*Then $g(x + \delta) = g(x), \forall \|\delta\|_1 \leq R$, where*

$$R = \max\left\{\frac{\lambda}{2}\log(\underline{P_A}/\overline{P_B}), -\lambda\log(1 - \underline{P_A} + \overline{P_B})\right\} \tag{1}$$

Some Remarks:

1. When $\underline{P_A} \to 1$ or $\overline{P_B} \to 0$, we can get $R \to \infty$. It is reasonable since the Laplace distribution is supported over $\mathbb{R}^d$, $\underline{P_A} \to 1$ is equivalent to $f = c_A$ almost everywhere.

2. Compared with (Lecuyer et al., 2019) where they have $R = \frac{\lambda}{2}\log(\underline{P_A}/\overline{P_B})$, our bound is better if $\frac{1-2\underline{P_A}(1-\underline{P_A})-\sqrt{1-4\underline{P_A}(1-\underline{P_A})}}{2\underline{P_A}} \leq \overline{P_B} \leq \frac{1-2\underline{P_A}(1-\underline{P_A})+\sqrt{1-4\underline{P_A}(1-\underline{P_A})}}{2\underline{P_A}}$. See Figure 4 for illustration, where we use baseline to denote the bound $R = \frac{\lambda}{2}\log(\underline{P_A}/\overline{P_B})$.

*Proof sketch:* (The full proof is in Appendix B) For arbitrarily classifier $f$, we can transform it into a random smoothing classifier $g$ using random smoothing technique, where $g$ returns class $c_A$ with probability no less than $\underline{P_A}$, and class $c_B$ with probability no more than $\overline{P_B}$. Below we list the three main ideas we used in our proof:

*1. How to deal with an arbitrary $f$ with $\underline{P_A}$ and $\overline{P_B}$?*

Following Cohen et al. (2019), we use Neyman-Pearson Lemma to transform the relation between $\mathbb{P}(f(X) = c_A)$ and $\mathbb{P}(f(Y) = c_A)$ into the relation between $\mathbb{P}(X \in A)$ and $\mathbb{P}(Y \in A)$. From Lemma 3.1, Neyman-Pearson Lemma still holds for mixed random variables.

*2. How to deal with the relation between $X = \epsilon$ and $Y = \epsilon + \delta$?*

Inspired by Lecuyer et al. (2019), we use the DP-form inequality ($P(Y \in A) \leq e^\epsilon P(X \in A)$) to deal with the relation between $P(X \in A)$ and $P(Y \in A)$. In Laplace distribution, $\epsilon = \frac{\|\delta\|_1}{\lambda}$.

*3. Take complements to get tighter bound.*

When $P_{A \text{ or } B} < 1/2$, the above DP-form inequality gets tighter. Therefore, we analyze $A^c$ when $\underline{P_A} \geq 1/2$ to get a new bound, and compare it with the baseline expression.

We derive this bound by Neyman-Pearson Lemma in this work, but an alternative approach is using Rényi Divergence (Li et al., 2018).

### 4.2 TIGHT ROBUSTNESS CERTIFICATES FOR BINARY CASE

Although we get improved result over Lecuyer et al. (2019), the bound in Theorem 1 is not tight since it considers the general case with multiple categories. In this section, we first present our result for binary classification (Theorem 2), which further improves over Theorem 1.

**Theorem 2** *(binary case) Let $f : \mathbb{R}^d \to Y$ be deterministic or random function, and let $\epsilon \sim \mathcal{L}(\lambda)$. Let $g(\boldsymbol{x}) = \arg\max_c \mathbb{P}_\epsilon(f(\boldsymbol{x} + \epsilon) = c)$. Suppose there are only two classes $c_A$ and $c_B$, and $\underline{P_A} \in [\frac{1}{2}, 1]$ s.t.*

$$\mathbb{P}(f(\boldsymbol{x} + \epsilon) = c_A) \geq \underline{P_A}$$

*Then $g(\boldsymbol{x} + \boldsymbol{\delta}) = g(\boldsymbol{x}), \forall \|\boldsymbol{\delta}\|_1 \leq R$, for*

$$R = -\lambda \log[2(1 - \underline{P_A})] \tag{2}$$

*Scretch of the proof:* (The full proof is in Appendix C) Theorem 2 is a special binary case of Theorem 1. We can use a method similar to Theorem 1 to get the results. However, it is worth noting that in binary cases, our new improved bound in Theorem 1 always dominates the bound by Lecuyer et al. (2019). Moreover, our bound in Eqn. (2) is tight, as shown below.

**Theorem 3** *(tight bound in binary case) In the same setting as Theorem 2, assume $\underline{P_A} + \overline{P_B} \leq 1$ and $\underline{P_A} \geq \frac{1}{2}$. $\forall R' > -\lambda \log[2(1 - \underline{P_A})]$, $\exists$ base classifier $f^*$ and perturbation $\boldsymbol{\delta}^*$ with $g^*(\boldsymbol{x}) = \arg\max_c \mathbb{P}_\epsilon(f^*(\boldsymbol{x} + \epsilon) = c)$ and $\|\boldsymbol{\delta}\|_1 = R'$, s.t. $g^*(\boldsymbol{x}) \neq g^*(\boldsymbol{x} + \boldsymbol{\delta}^*)$.*

*Scretch of the proof:*(The full proof is in Appendix C) For Theorem 3, we prove that the bound in Theorem 2 is tight by calculating the results in one-dimensional case, where $\boldsymbol{\delta} = (\|\boldsymbol{\delta}\|_1, 0, \dots, 0)$.

By calculating, we show that when $\boldsymbol{\delta} = (\|\boldsymbol{\delta}\|_1, 0, \dots, 0)$

$$\mathbb{P}(Y \in B) = \int_{-\infty}^{\|\boldsymbol{\delta}\|_1 + \lambda \log[2\overline{P_B}]} \frac{1}{2\lambda} \exp\left(-\frac{|x|}{\lambda}\right) dx$$

$$= \begin{cases} \exp(\frac{\|\boldsymbol{\delta}\|_1}{\lambda})\overline{P_B} & \text{when } \|\boldsymbol{\delta}\|_1 \leq -\lambda \log[2\overline{P_B}] \\ 1 - \frac{1}{4\overline{P_B}} \exp(-\frac{\|\boldsymbol{\delta}\|_1}{\lambda}) & \text{o.w.} \end{cases}$$

Therefore, when $\|\boldsymbol{\delta}\|_1 \leq -\lambda \log[2\overline{P_B}]$, the DP-inequality is tight. The worst-case $\boldsymbol{\delta}$ appears in the one-dimension case.

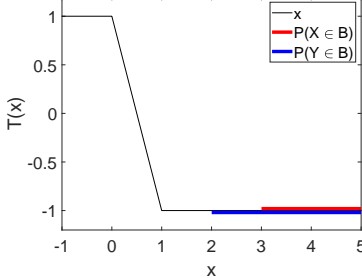

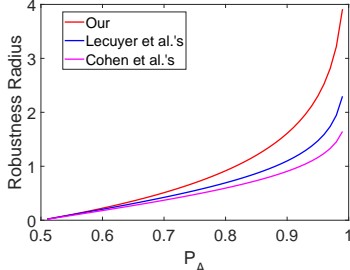

Figure 5: When $\boldsymbol{\delta}$ is small, we will take the red part to construct $\mathbb{P}(X \in B)$, and blue part to construct $\mathbb{P}(Y \in B)$. The difference between them meets the condition that $T(x) = -\|\boldsymbol{\delta}\|_1$, which leads to a tight bound.

Figure 6: Comparing different methods under different $\underline{P_A}$. Our model always gives the largest radius compared with the other models, because our bound is tight.

Figure 5 shows the reason why the inequality is tight. When $\boldsymbol{\delta}$ is small, for $\mathbb{P}(X \in B)$, the set $B$ we selected satisfies $\forall \boldsymbol{x} \in B, T(\boldsymbol{x}) = -\|\boldsymbol{\delta}\|_1$ (red part). When $\mathbb{P}(Y \in B)$ is considered, it moves set $S$ towards left by step $\boldsymbol{\delta}$. However, due to the small $\delta$, $S$ after moving still satisfies the requirement of $\forall \boldsymbol{x} \in S, T(\boldsymbol{x}) = -\|\boldsymbol{\delta}\|_1$ (blue part). Therefore, the inequality is tight.

### 4.3 METHOD COMPARISON

We compared our method with Cohen et al.'s and Lecuyer et al.'s in binary case, see Table 1. We plot the curves in Figure 6. As we can see, under the same variance of each noise, our method can reach better robustness radius. Below we show simple derivations of the bounds in Table 1.

Table 1: Robustness Radius Comparison

| Method | Noise | Radius |
|---|---|---|
| Our | Laplace $L(0, \lambda)$ | $-\lambda \log[2(1 - \underline{P_A})]$ |
| Lecuyer et al.'s | Laplace $L(0, \lambda)$ | $\frac{\lambda}{2} \log(\underline{P_A}/1 - \underline{P_A})$ |
| Cohen et al.'s | Gaussian $N(0, \sigma^2)$ | $\sigma \Phi^{-1}(\underline{P_A})$ |

**Robustness radius of Lecuyer et al. (2019)**

Using the basic inequality from differential privacy, we have:

$$\mathbb{P}\left(f(X) = c_A\right) \leq \exp(\beta)\mathbb{P}\left(f(Y) = c_A\right)$$
$$\mathbb{P}\left(f(Y) = c_B\right) \leq \exp(\beta)\mathbb{P}\left(f(X) = c_B\right)$$

where $\beta = \|\boldsymbol{\delta}\|_1/\lambda$. The above two inequalities show that to guarantee $\mathbb{P}\left(f(Y) = c_A\right) > \mathbb{P}\left(f(Y) = c_B\right)$, it suffices to show that:

$$\mathbb{P}\left(f(X) = c_A\right) > \exp(2\beta)\mathbb{P}\left(f(X) = c_B\right)$$

So plug in $\beta = \|\boldsymbol{\delta}\|_1/\lambda$, we have $\|\boldsymbol{\delta}\|_1 \leq \frac{\lambda}{2} \log(\underline{P_A}/\overline{P_B})$. Furthermore, in binary case, we can plug in $\overline{P_B} = 1 - \underline{P_A}$, and get the robustness radius: $R = \frac{\lambda}{2} \log(\underline{P_A}/1 - \underline{P_A})$.

**Robustness radius of Cohen et al. (2019)**

Denote $\mathcal{B}_{p,r}(c) = \{x : \|x - c\|_p \leq r\}$. Since we know that $\mathcal{B}_{1,r}(c) \subset \mathcal{B}_{2,r}(c)$, so the radius in (Cohen et al., 2019) can be directly used in $\ell_1$ form, which is $\sigma \Phi^{-1}(\underline{P_A})$.

Besides, since $\mathcal{B}_{1,r+\epsilon}(c) \not\subset \mathcal{B}_{2,r}(c)$ whatever $\epsilon > 0$ is. And (Cohen et al., 2019) is an exact robustness guarantee, so we have that the best $\ell_1$ form that isotropic Gaussian noise random smoothing can get is $\sigma \Phi^{-1}(\underline{P_A})$.

Finally we will prove that $-\lambda \log[2(1 - \underline{P_A})] \geq \frac{\lambda}{2} \log(\underline{P_A}/1 - \underline{P_A})$. For simple denotion, we just set $\underline{P_A} = p \geq 0.5$. So it is sufficient to show that $-\lambda \log[2(1-p)] \geq \frac{\lambda}{2} \log(p/(1-p))$. By applying exponential operation, it suffices to show that $\frac{1}{2(1-p)} \geq \sqrt{\frac{p}{1-p}}$, which is simply $p(1-p) \leq \frac{1}{4}$.

## 5 EXPERIMENTS

### 5.1 IMPLEMENTATION DETAILS

**Monte Carlo.** Since we cannot get the exact value of $P_A$, we have to use Monte Carlo method to get the approximate value of $P_A$. More specifically, we take multiple random samples from the Laplace distribution to estimate $\underline{P_A}$. One way to do it is grouping the samples and get $\underline{P_A}$ using non-parametric estimation.

In our experiments, we applied two different types of training, as described below.

**Type1-Training:** The first method is intuitive, and was applied in (Cohen et al., 2019). In the training process, we add into inputs:

$$\text{inputs} = \text{inputs} + \text{noise}$$

where the noise is sampled from isotropic Laplace distribution.

**Type2-Training:** The second method was recently proposed by Salman et al. (2019). The idea is to use *adversarial noise samples* instead of the *raw noise samples* in a neighborhood to train the base classifier. Each training sample can be decomposed to

$$\text{inputs} = \text{inputs} + \text{noise} + \text{perturbation}$$

where the noise comes from an isotropic Laplace distribution, and the perturbation is found approximately by the gradient of loss with respect to the input. Concretely, if we denote the loss as $L$ and the input as $x$, the perturbation $\Delta$ can be calculated by $\Delta = a * \text{sign}(\nabla_x L(\theta, x, y))$, where $a$ is a constant.

**Evaluation Index.** In this paper, we choose certified accuracy as our evaluation index. Robustness certified accuracy at radius $r$ refers to the proportion of correctly classified samples with at least robustness radius $r$. Specifically, if a group of samples with capacity $n$ is $\{x_i\}, i = 1, 2, \ldots, n$, its corresponding certified robustness radius is $R_i$. An index $x_i$ represent if the sample is classified correctly. If the sample is correctly classified, $x_i = 1$, otherwise $x_i = 0$. For a given $r$, the corresponding robustness certified accuracy is defined as $\alpha = \sum_{i=1}^{n} x_i \mathbb{1}(R_i \geq r)/n$, where $\mathbb{1}(\cdot)$ is an indicator function.

However, from Section 5.1 we know that we cannot calculate the exact robustness radius $R$, so we use its $\hat{R}$ to approximate $R$, which leads to a "approximate robustness certified accuracy"($\hat{\alpha}$), which is calculated by

$$\hat{\alpha} = \sum_{i=1}^{n} x_i I(\hat{R}_i \geq r)/n \qquad (3)$$

Cohen et al. (2019) demonstrates that when significance level of $\hat{R}$ is small, the difference between these two quantities is negligible. In practice, we plot approximate certified accuracy $\hat{\alpha}$ as a function of radius $r$. From Eqn. (3), we know that $\hat{\alpha}$ is non-increasing w.r.t. $r$. And when $r \to \infty$, $\hat{\alpha} \to 0$.

**Hyperparameters.** In our paper, we set all our hyperparameters following Cohen et al. (2019). Specifically, we set significance level to 0.001. $n_0 = 100$ in Monte Carlo simulation (used to get bound for $\hat{\alpha}$) and $n = 100,000$ in estimation part (used to estimate $\hat{\alpha}$). Moreover, we test three parameters in CIFAR-10 dataset and ImageNet dataset ($\sigma = 0.25, 0.50, 1.00$). Since (Cohen et al., 2019) use Gaussian noise and we use Laplace noise, they should have the same standard deviation during comparison. This requires that $\sigma = \sqrt{2}\lambda$.

## 5.2 EXPERIMENTAL RESULTS

**Results on ImageNet and CIFAR-10.** We applied random smoothing on CIFAR-10 (Krizhevsky (2009)) and ImageNet (Deng et al. (2009)) respectively. On each data set, we trained several random smoothing models with differential standard deviation $\sigma$ for Laplace noise. In order to keep in line with Cohen et al.'s method and make a comparison, we select $\sigma = 0.25, 0.50, 1.00$ on CIFAR-10, and ImageNet, corresponding parameter $\lambda = \sigma/\sqrt{2}$.

Figure 6 draws the certified accuracy achieved by smoothing with each sigma. For the ImageNet dataset, we only use the most basic training method (Type1 Training). For the CIFAR-10 data set, we use two training methods (Type 1 and Type 2 Training). We can see that the smaller sigma performs better when the radius is smaller. As the noise gets bigger, the accuracy becomes lower, but the robustness guarantee becomes higher. The dashed black line shows the empirical robust accuracy of an undefended classifier from Cohen et al. (2019).

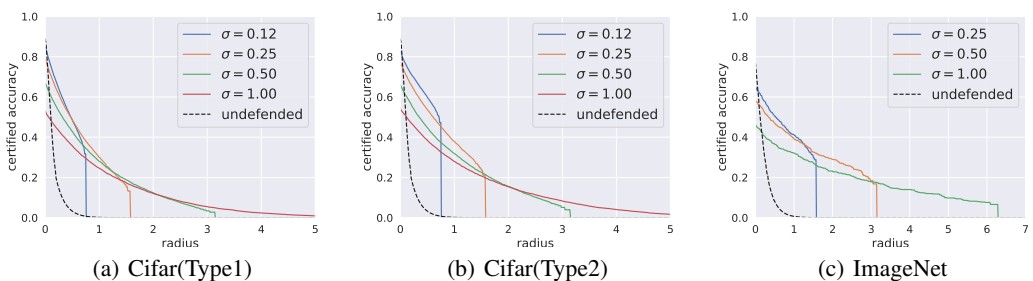

(a) Cifar(Type1)          (b) Cifar(Type2)          (c) ImageNet

Figure 6: Approximate certified accuracy on CIFAR-10 and ImageNet.

**Comparison with baseline.**

We will show our comparison results in the following. Based on Table. 1, we will test our method on CIFAR-10 with the ResNet110 architecture as well as Type1 and Type2 training, and ImageNet with ResNet50 architecture as well as Type1 training. We will compare our results with (Cohen et al., 2019) and (Lecuyer et al., 2019) under the same standard deviation $\sigma$. For base classifiers, ours and Lecuyer et al.'s share the same base classifier with Laplace training noise, and Cohen et al.'s uses the base classifier with Gaussian training noise.

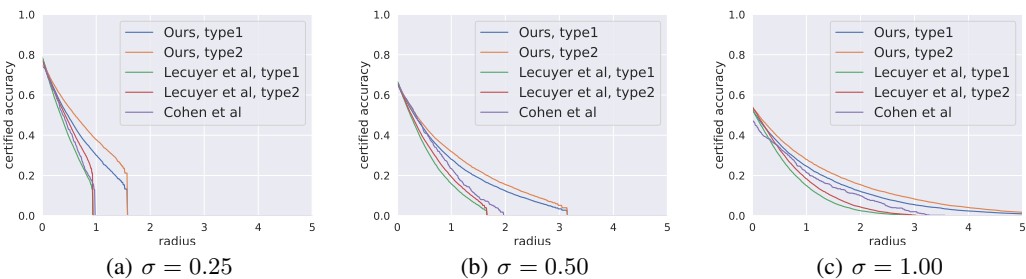

(a) $\sigma = 0.25$          (b) $\sigma = 0.50$          (c) $\sigma = 1.00$

Figure 7: Approximate certified accuracy attained by randomized smoothing on CIFAR-10.

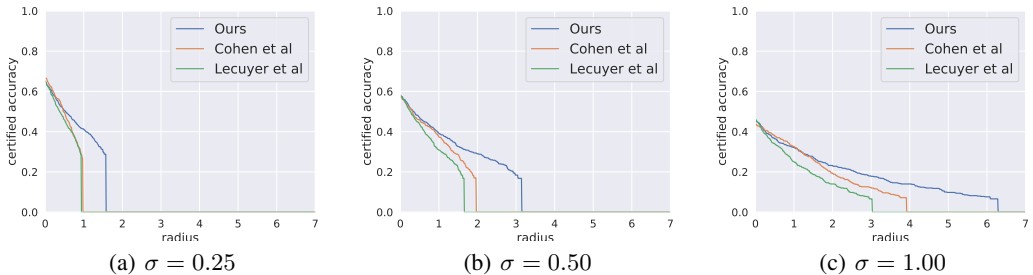

(a) $\sigma = 0.25$          (b) $\sigma = 0.50$          (c) $\sigma = 1.00$

Figure 8: Approximate certified accuracy attained by randomized smoothing on ImageNet.

## 6   CONCLUSION

In this paper, we combine the inequality from differential privacy and the classic Neyman-Pearson Lemma to resolve the challenging asymmetry of $\ell_1$ metric and the mixed discrete-continuous property of the likelihood ratios under isotropic Laplace distributions. In addition, by comparing the high-dimensional case with a special edge case, we prove the tight $\ell_1$ robustness guarantee for binary classification problems, and obtain the state-of-the-art certified accuracy in large scale experiments.

The establishment of $\ell_1$ certificate via Laplace distributions and the prior result of $\ell_2$ certificate via Gaussian distributions may be extended to a generic theorem for a general $\ell_p$ norm robustness certificate via the associated realization of the generalized Gaussian distribution, where the aforementioned results are special cases of the general scheme. The introduction of the mixed random variable analysis and $\ell_p$ geometry analysis may serve as a valuable extension of existing works towards such general goal.

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

# A  PROOF OF LEMMA 1

In this section, we will prove that Neyman-Pearson Lemma holds with mixed random variable.

WLOG, $\boldsymbol{x} = \boldsymbol{0}$, $X \sim \mathcal{L}(\lambda)$ and $Y \sim \mathcal{L}(\lambda) + \boldsymbol{\delta}$. We will firstly introduce Neyman-Pearson Lemma, which plays an important role in our proof.

**Lemma 3.1 (restated)**. Let $X \sim \mathcal{L}(\lambda)$ and $Y \sim \mathcal{L}(\lambda) + \boldsymbol{\delta}$. Let $h : \mathbb{R}^d \to \{0, 1\}$ be any deterministic or random function. Given any $\beta \in \mathbb{R}$, and $S' \subseteq \{\boldsymbol{z} \in \mathbb{R}^d : \|\boldsymbol{z} - \boldsymbol{\delta}\|_1 - \|\boldsymbol{z}\|_1 = \beta\}$:

1. If $S = \{\boldsymbol{z} \in \mathbb{R}^d : \|\boldsymbol{z} - \boldsymbol{\delta}\|_1 - \|\boldsymbol{z}\|_1 > \beta\} \cup S'$, and $\mathbb{P}(h(X) = 1) \geq \mathbb{P}(X \in S)$ then $\mathbb{P}(h(Y) = 1) \geq \mathbb{P}(Y \in S)$

2. If $S = \{\boldsymbol{z} \in \mathbb{R}^d : \|\boldsymbol{z} - \boldsymbol{\delta}\|_1 - \|\boldsymbol{z}\|_1 < \beta\} \cup S'$, and $\mathbb{P}(h(X) = 1) \leq \mathbb{P}(X \in S)$, then $\mathbb{P}(h(Y) = 1) \leq \mathbb{P}(Y \in S)$

*Proof of Lemma 3.1*  First, notice that $\mathbb{P}(X \in S)$ can be regarded as a mixed random variable. We want to prove that as long as we can choose a $S'$ that satisfies $\mathbb{P}(X \in S) \leq \mathbb{P}(h(X) = 1)$, Neyman-Pearson Lemma can always hold.

Let's first see what happens in the proof of Neyman-Pearson Lemma. Notice that $X$ and $Y$ are continuous variables, but $X \in S$ and $Y \in S$ can be regarded as mixed continuous-discrete event. Then we can choose a reasonable $S'$ for $X$ and $Y$. We will prove case 1 and the other one can be proved with similar method.

$$
\begin{aligned}
&\mathbb{P}(h(Y) = 1) - \mathbb{P}(Y \in S) \\
=& \int_{\mathbb{R}^d} h(1|z)\mu_Y(z)\mathrm{d}z - \int_S \mu_Y(z)\mathrm{d}z \\
=& [\int_{S^c} h(1|z)\mu_Y(z)\mathrm{d}z + \int_S h(1|z)\mu_Y(z)\mathrm{d}z] - [\int_S h(1|z)\mu_Y(z)\mathrm{d}z + \int_S h(0|z)\mu_Y(z)\mathrm{d}z] \\
=& \int_{S^c} h(1|z)\mu_Y(z)\mathrm{d}z - \int_S h(0|z)\mu_Y(z)\mathrm{d}z \\
\geq& t(\int_{S^c} h(1|z)\mu_X(z)\mathrm{d}z - \int_S h(0|z)\mu_X(z)\mathrm{d}z) \\
=& t([\int_{S^c} h(1|z)\mu_X(z)\mathrm{d}z + \int_S h(1|z)\mu_X(z)\mathrm{d}z] - [\int_S h(0|z)\mu_X(z)\mathrm{d}z + \int_{S^c} h(1|z)\mu_X(z)\mathrm{d}z]) \\
=& t(\mathbb{P}(h(X) = 1) - \mathbb{P}(X \in S)) \\
\geq& 0
\end{aligned}
\tag{4}
$$

The first inequality holds due to the construction of mixed definition $S$. If $\boldsymbol{z} \in S$, $\frac{\mu_Y(\boldsymbol{z})}{\mu_X(\boldsymbol{z})} \geq t$. If $\boldsymbol{z} \in S^c$, $\frac{\mu_Y(\boldsymbol{z})}{\mu_X(\boldsymbol{z})} \leq t$. Compared with continuous set, the only difference appears in the equal sign.

It should be noted that $\mathbb{P}(X \in S)$ and $\mathbb{P}(Y \in S)$ should keep consistent, which means that they should have the same $S'$. In this derivation, we can find that $P(X \in S)$ and $P(Y \in S)$ use the same set $S'$ in Eqn. (4).

Next, we will plug in the condition that X and Y are isotropic Laplaces.

Then we just need to prove that

$$
\left\{\boldsymbol{z} \in \mathbb{R}^d : \frac{\mu_Y(\boldsymbol{z})}{\mu_X(\boldsymbol{z})} \leq t\right\} \Longleftrightarrow \left\{\boldsymbol{z} \in \mathbb{R}^d : \|\boldsymbol{z} - \boldsymbol{\delta}\|_1 - \|\boldsymbol{z}\|_1 \geq \beta\right\}
$$

When X and Y are isotropic Laplaces, the likelihood ratio turns out to be:

$$
\begin{aligned}
\frac{\mu_Y(\boldsymbol{z})}{\mu_X(\boldsymbol{z})} &= \frac{\exp(-\frac{1}{\lambda}\|\boldsymbol{z} - \boldsymbol{\delta}\|_1)}{\exp(-\frac{1}{\lambda}\|\boldsymbol{z}\|_1)} \\
&= \exp(-\frac{1}{\lambda}(\|\boldsymbol{z} - \boldsymbol{\delta}\|_1 - \|\boldsymbol{z}\|_1))
\end{aligned}
$$

By choosing $\beta = -\lambda \log(t)$, we can derive that

$$\|z - \delta\|_1 - \|z\|_1 \geq \beta \iff \frac{\mu_Y(z)}{\mu_X(z)} \leq t$$

$$\|z - \delta\|_1 - \|z\|_1 \leq \beta \iff \frac{\mu_Y(z)}{\mu_X(z)} \geq t$$

## B    PROOF OF THEOREM 1

**Theorem 1(restated)** Let $f : \mathbb{R}^d \to Y$ be deterministic or random function, and let $\epsilon \sim \mathcal{L}(\lambda)$. Let $g(\boldsymbol{x}) = \arg\max_c \mathbb{P}_\epsilon(f(\boldsymbol{x} + \epsilon) = c)$. Suppose $\underline{P_A}, \overline{P_B} \in [0, 1]$ are such that
$$\mathbb{P}\left(f(\boldsymbol{x} + \epsilon) = c_A\right) \geq \underline{P_A} \geq \overline{P_B} \geq \max_{c \neq c_A} \mathbb{P}(f(\boldsymbol{x} + \epsilon) = c)$$
Then $g(\boldsymbol{x} + \boldsymbol{\delta}) = g(\boldsymbol{x}), \forall \|\boldsymbol{\delta}\|_1 \leq R$, where

$$R = \max\left\{\frac{\lambda}{2}\log(\underline{P_A}/\overline{P_B}), -\lambda \log(1 - \underline{P_A} + \overline{P_B})\right\} \tag{5}$$

*Proof of Theorem 1*    Denote $T(\boldsymbol{x}) = \|\boldsymbol{x} - \boldsymbol{\delta}\|_1 - \|\boldsymbol{x}\|_1$. Use Triangle Inequality we can derive a bound for $T(x)$:
$$-\|\boldsymbol{\delta}\|_1 \leq T(\boldsymbol{x}) \leq \|\boldsymbol{\delta}\|_1 \tag{6}$$
Pick $\beta_1, \beta_2$ such that there exists $A' \subseteq \{\boldsymbol{z} : T(\boldsymbol{z}) = \beta_1\}, B' \subseteq \{\boldsymbol{z} : T(\boldsymbol{z}) = \beta_2\}$, and
$$\mathbb{P}(X \in \{\boldsymbol{z} : T(\boldsymbol{z}) > \beta_1\} \cup A') = \underline{P_A} \leq \mathbb{P}(f(X) = c_A))$$
$$\mathbb{P}(X \in \{\boldsymbol{z} : T(\boldsymbol{z}) < \beta_2\} \cup B') = \overline{P_B} \geq \mathbb{P}(f(X) = c_B)$$
Define
$$A := \{\boldsymbol{z} : T(\boldsymbol{z}) > \beta_1\} \cup A'$$
$$B := \{\boldsymbol{z} : T(\boldsymbol{z}) < \beta_2\} \cup B'$$

Thus, apply Lemma 3.1, we have
$$\begin{aligned}\mathbb{P}(Y \in A) &\leq \mathbb{P}(f(Y) = c_A) \\ \mathbb{P}(Y \in B) &\geq \mathbb{P}(f(Y) = c_B)\end{aligned} \tag{7}$$

Then consider $\mathbb{P}(Y \in A)$ and $\mathbb{P}(Y \in B)$

$$\begin{aligned}\mathbb{P}(Y \in A) &= \int_A [2\lambda]^{-d} \exp(-\frac{\|\boldsymbol{x} - \boldsymbol{\delta}\|_1}{\lambda})\mathrm{dx} \\ &= \int_A [2\lambda]^{-d} \exp(-\frac{\|\boldsymbol{x}\|_1}{\lambda}) \exp(-\frac{T(\boldsymbol{x})}{\lambda})\mathrm{dx} \\ &\geq \exp(-\frac{\|\boldsymbol{\delta}\|_1}{\lambda}) \int_A [2\lambda]^{-d} \exp(-\frac{\|x\|_1}{\lambda})\mathrm{dx} \\ &= \exp(-\frac{\|\boldsymbol{\delta}\|_1}{\lambda})\underline{P_A}\end{aligned} \tag{8}$$

The inequality is derived by Eqn.( 6). Similarly, we can get

$$\begin{aligned}\mathbb{P}(Y \in B) &= \int_B [2\lambda]^{-d} \exp(-\frac{\|\boldsymbol{x} - \boldsymbol{\delta}\|_1}{\lambda})\mathrm{dx} \\ &= \int_B [2\lambda]^{-d} \exp(-\frac{\|\boldsymbol{x}\|_1}{\lambda}) \exp(-\frac{T(\boldsymbol{x})}{\lambda})\mathrm{dx} \\ &\leq \exp(\frac{\|\boldsymbol{\delta}\|_1}{\lambda}) \int_B [2\lambda]^{-d} \exp(-\frac{\|\boldsymbol{x}\|_1}{\lambda})\mathrm{dx} \\ &= \exp(\frac{\|\boldsymbol{\delta}\|_1}{\lambda})\overline{P_B}\end{aligned} \tag{9}$$

First, we would like to show that **robustness can be guaranteed when** $R \leq \frac{\lambda}{2}\log(\underline{P_A}/\overline{P_B})$**.**

If $\|\boldsymbol{\delta}\|_1 \leq \frac{\lambda}{2}\log(\underline{P_A}/\overline{P_B})$, by Eqn. (7, 8, 9), we have
$$\mathbb{P}(f(Y) = c_A) \geq \mathbb{P}(Y \in A) \geq \mathbb{P}(Y \in B) \geq \mathbb{P}(f(Y) = c_B)$$

Then, we would like to show that **robustness can be guaranteed when** $R \leq -\lambda\log(1 - \underline{P_A} + \overline{P_B})$**.**

From Eqn. (9), we know that $\mathbb{P}(Y \in B) \leq \exp(\frac{\|\boldsymbol{\delta}\|_1}{\lambda})\overline{P_B}$. Besides, by applying Eqn. (9) in set $A^c$, we can get that $\mathbb{P}(Y \in A) \geq 1 - \exp(\frac{\|\boldsymbol{\delta}\|_1}{\lambda})(1 - \underline{P_A})$. So we can calculate that if $\|\boldsymbol{\delta}\|_1 \leq -\lambda\log(1 - \underline{P_A} + \overline{P_B})$, we have
$$\mathbb{P}(f(Y) = c_A) \geq \mathbb{P}(Y \in A) \geq \mathbb{P}(Y \in B) \geq \mathbb{P}(f(Y) = c_B)$$

Moreover, by simple algebraic operation, we can derive that $-\lambda\log(1 - \underline{P_A} + \overline{P_B}) \geq \frac{\lambda}{2}\log(\underline{P_A}/\overline{P_B})$ requires $\frac{1 - 2\underline{P_A}(1 - \underline{P_A}) - \sqrt{1 - 4\underline{P_A}(1 - \underline{P_A})}}{2\underline{P_A}} \leq \overline{P_B} \leq \frac{1 - 2\underline{P_A}(1 - \underline{P_A}) + \sqrt{1 - 4\underline{P_A}(1 - \underline{P_A})}}{2\underline{P_A}}$.

The proof for Theorem 1 is finished.

## C  PROOF OF THEOREM 2 AND THEOREM 3

**Theorem 2(restated) (binary case)** Let $f : \mathbb{R}^d \to Y$ be deterministic or random function, and let $\boldsymbol{\epsilon} \sim \mathcal{L}(\lambda)$. Let $g(\boldsymbol{x}) = \arg\max_c \mathbb{P}_{\boldsymbol{\epsilon}}(f(\boldsymbol{x} + \boldsymbol{\epsilon}) = c)$. Suppose there are only two classes $c_A$ and $c_B$, and $\underline{P_A} \in [\frac{1}{2}, 1]$ s.t.
$$\mathbb{P}(f(\boldsymbol{x} + \boldsymbol{\epsilon}) = c_A) \geq \underline{P_A}$$
Then $g(\boldsymbol{x} + \boldsymbol{\delta}) = g(\boldsymbol{x}), \forall \|\boldsymbol{\delta}\|_1 \leq R$, for

$$R = -\lambda\log[2(1 - \underline{P_A})] \tag{10}$$

*Proof of Theorem 2:*

It is similar to the proof of Theorem 1. Pick $\beta_3$ such that there exists $B' \subseteq \{\boldsymbol{z} : T(\boldsymbol{z}) = \beta_3\}$, and
$$\mathbb{P}(X \in \{\boldsymbol{z} : T(\boldsymbol{z}) < \beta_3\} \cup B') = \overline{P_B} = \mathbb{P}(f(X) = c_B)$$

Define
$$S := \{\boldsymbol{z} : T(\boldsymbol{z}) < \beta_3\} \cup B'$$

So we also have $\mathbb{P}(X \notin S) = P_A = \mathbb{P}(f(X) = c_A)$. Plug into Lemma 3.1, we can get
$$\mathbb{P}(Y \notin S) \leq \mathbb{P}(f(Y) = c_A)$$
$$\mathbb{P}(Y \in S) \geq \mathbb{P}(f(Y) = c_B)$$

Using a similar method as Eqn. (9), we can get that
$$\mathbb{P}(Y \in S) \leq \exp(\frac{\|\boldsymbol{\delta}\|_1}{\lambda})P_B$$

Since we have
$$P_B = \mathbb{P}(f(X) = c_B) = 1 - P_A \leq 1 - \underline{P_A}$$

Thus, if $\|\boldsymbol{\delta}\|_1 \leq R = -\lambda\log[2(1 - \overline{P_A})]$, it holds that
$$\mathbb{P}(Y \in S) \leq \exp(\frac{\|\boldsymbol{\delta}\|_1}{\lambda})P_B$$
$$\leq \exp(\frac{-\lambda\log[2(1 - \overline{P_A})]}{\lambda})(1 - \underline{P_A})$$
$$= \frac{1}{2}$$

That is to say, $\mathbb{P}(f(Y) = c_A) \geq \mathbb{P}(Y \notin S) \geq \frac{1}{2} \geq \mathbb{P}(Y \in S) \geq \mathbb{P}(f(Y) = c_B)$.

The proof for Theorem 2 is finished.

**Theorem 3(restated) (tight bound in binary case)** In the same setting as Theorem 2, assume $\underline{P_A} + \overline{P_B} \leq 1$ and $\underline{P_A} \geq \frac{1}{2}$. $\forall R' > -\lambda \log[2(1 - \underline{P_A})]$, $\exists$ base classifier $f^*$ and perturbation $\boldsymbol{\delta}^*$ with $g^*(\boldsymbol{x}) = \arg\max_c \mathbb{P}_\epsilon(f^*(\boldsymbol{x} + \boldsymbol{\epsilon}) = c)$ and $\|\boldsymbol{\delta}\|_1 = R'$, s.t. $g^*(\boldsymbol{x}) \neq g^*(\boldsymbol{x} + \boldsymbol{\delta}^*)$.

*Proof of Theorem 3:* Here, we first set $\boldsymbol{\delta} = (\|\boldsymbol{\delta}\|_1, 0, \ldots, 0)$. For simplification, we denote $\overline{\delta} = \|\boldsymbol{\delta}\|_1$. And define

$$A := \left\{ z : |z - \overline{\delta}| - |z| \geq \max\{\overline{\delta} + 2\lambda \log\left[2\left(1 - \underline{P_A}\right)\right], -\overline{\delta}\} \right\}$$

Then, we can calculate that

$$
\begin{aligned}
\mathbb{P}(X \in A) &= \mathbb{P}_x(|x - \overline{\delta}| - |x| \geq \max\{\overline{\delta} + 2\lambda \log[2(1 - \underline{P_A})], -\overline{\delta}\}) \\
&= \int_{-\infty}^{-\lambda \log[2(1-\underline{P_A})]} \frac{1}{2\lambda} \exp\left(-\frac{|x|}{\lambda}\right) dx \\
&= 1 - \int_{-\lambda \log[2(1-\underline{P_A})]}^{\infty} \frac{1}{2\lambda} \exp\left(\frac{x}{\lambda}\right) dx \\
&= \underline{P_A}
\end{aligned}
\tag{11}
$$

where $x \sim \frac{1}{2\lambda} \exp\left(-\frac{|x|}{\lambda}\right)$, $\overline{\delta} = \|\boldsymbol{\delta}\|_1$. Notice that if $\overline{\delta} + 2\lambda \log[2(1 - \underline{P_A})] \leq -\overline{\delta}$, we will get the integral equation by choosing $S'$. With Eqn. (11), we have

$$\mathbb{P}(X \in A) = \underline{P_A} \leq \mathbb{P}(f(X) = c_A) \tag{12}$$

Thus, plug Eqn. (12) into the results of Lem. 3.1, we have

$$\mathbb{P}(Y \in A) \leq \mathbb{P}(f(Y) = c_A) \tag{13}$$

Also, since $Y = X + \overline{\delta}$, it can be derived that

$$\mathbb{P}(Y \in A) = \int_{-\infty}^{-\lambda \log[2(1-\underline{P_A})]-\overline{\delta}} \frac{1}{2\lambda} \exp\left(-\frac{|x|}{\lambda}\right) dx \tag{14}$$

Here we use the consistency of $X \in A$ and $Y \in A$. Since $Y$ can be regarded as an offset of $X$, the integral limit should be translated into the same length. So, if $\|\boldsymbol{\delta}\|_1 = \overline{\delta} \leq -\lambda \log[2(1 - \underline{P_A})]$, by Eqn. (7) and Eqn. (14), we have

$$\mathbb{P}(f(Y) = c_A) \geq \mathbb{P}(Y \in A) \geq \frac{1}{2}$$

This means that the results we get in binary case is a tight bound, and the worst-case $\boldsymbol{\delta}$ appears when $\boldsymbol{\delta} = (\overline{\delta}, 0, \ldots, 0)$. Furthermore, if we slightly enlarge $\overline{\delta}$, there would be a case that the robustness is destroyed.

The proof for Theorem 3 is finished.

## D   WHY LAPLACE NOISE INSTEAD OF GAUSSIAN

In this section, we theoretically analyze the certification capabilities of Gaussian and Laplace noises. We will show that, given the same base classifier $f$ the parameter of Laplace distributions $\lambda$ is less sensitive than the parameter of Gaussian distributions $\sigma$. Given a base classifier $f$, where

$$f(x) = \begin{cases} c_A & |x| \leq 1 \\ c_B & o.w. \end{cases}$$

and two random smoothing functions

$$g_1(x) = \arg\max_c \mathbb{P}(f(x + \epsilon) = c), \epsilon \sim \mathcal{L}(0, \lambda),$$

$$g_2(x) = \arg\max_c \mathbb{P}(f(x + \epsilon) = c), \epsilon \sim \mathcal{N}(0, \sigma^2),$$

we aim to prove that Laplace noises will better protect the original prediction than Gaussian noises.

Formally, we compare their **Rectified Optional Parameter Space (ROPS)**, defined as $\Lambda = \{\sqrt{2}\lambda : g_1(x; \lambda) = f(x)\}$ and $\Sigma = \{\sigma : g_2(x; \sigma) = f(x)\}$. Note that the rectified term $\sqrt{2}$ is due to the fact that $\sigma = \sqrt{2}\lambda$ yield the same variance. Essentially, ROPS indicates the feasible region where the smoothing distribution does not negatively impact the base classifier, thus measuring the sensitivity of the smoothing distribution (the larger the better).

First, we would like to compare its prediction on a given point $(x, f(x)) = (0, c_A)$. We have

$$g_1(0) = c_A \iff \mathbb{P}(f(0 + \epsilon) = c_A) \geq \frac{1}{2} \iff \mathbb{P}(|\epsilon| \leq 1) = 1 - \exp(-\frac{1}{\lambda}) \geq \frac{1}{2} \iff \lambda \leq \frac{1}{\log 2},$$

$$g_2(0) = c_A \iff \mathbb{P}(f(0 + \epsilon) = c_A) \geq \frac{1}{2} \iff \mathbb{P}(|\epsilon| \leq 1) = 2\Phi(\frac{1}{\sigma}) - 1 \geq \frac{1}{2} \iff \sigma \leq \frac{1}{\Phi^{-1}(3/4)}.$$

Since $\frac{\sqrt{2}}{\log 2} > \frac{1}{\Phi^{-1}(3/4)}$, Laplace noises have a larger ROPS than Gaussian noises at the point $x = 0$.

The analysis can be further extended in two cases.

First, if we have $x \neq 0$, what is the corresponding ROPS that leads to the desired result ($g(x) = f(x)$)? We show in Fig. 10 that we will have a larger ROPS under Laplace noises.

Second, if we have a fixed $x$ but fixed a desired certified radius, what is the corresponding ROPS? We show in Fig. 11 that Laplace noises again have a larger ROPS.

We empirically validate this finding with ResNet110 on CIFAR-10. The resulting smoothed model has 24.8% clean accuracy under a Laplace noise, and 23.7% clean accuracy under a Gaussian noise (with the same variance as the Laplace noise). Here the accuracy is computed with respect to predictions of the base classifier instead of the labels (to illustrate how the smoothing impacts the predictions).

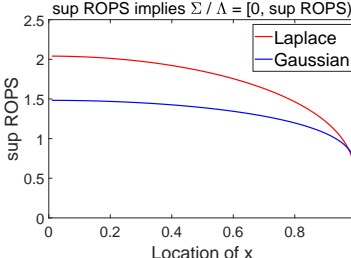

Figure 10: The ROPS under various $x$. Here $\Sigma = [0, \sup \text{ROPS})$, and similarly for $\Lambda$. Laplace noises are less sensitive than Gaussian noises in terms of ROPS.

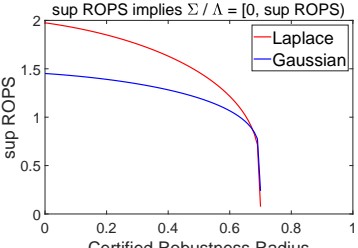

Figure 11: The ROPS under various certified robustness radii with a fixed $x = 0.3$ (other $x$ yields similar results). Laplace noises are less sensitive than Gaussian noises in terms of ROPS.

