# OpenReview forum: "$\ell_1$ Adversarial Robustness Certificates: a Randomized Smoothing Approach"
_ICLR.cc/2020/Conference — Reject_

### Official Review · AnonReviewer1 · 2019-10-22
**Official Blind Review #1**

**Rating:** 3

**Review:**

In this paper, the author derived a tight ell_1, which is not the symmetric norm, robustness certificates under isotropic Laplace distributions. Experimentally, the authors showed that the deep networks smoothed
by Laplace distributions yield the state-of-the-art certified robustness in ell_1 norm on the CIFAR-10
and ImageNet. To find the ell_1 certificate, the authors first identified the tight robustness certificate, for attacking the model in one particular direction, say the first direction. To show that any other perturbation directions cannot lead to a worse result, the authors convert the d dimensional likelihood function into a one-dimensional function, and the authors used relaxation for different perturbations and show that the worst-case result is bounded by the previously identified direction.  However, I have the following concerns about this work:

1. Theoretically, the authors only showed the certificate is tight for binary classification. I would suggest
the author change their claim in the abstract.

2. What is M on page 3 which is used without definition after definition 1?

3. Can you give a concrete continuous probability distribution that leads to the scenario in Fig.~3?

4. Can you extend the analysis to a multi-class classification scenario?

5. Besides randomized smoothing on the input images, recently Wang et al showed that randomize the deep nets can
also improve the deep nets and they gave it a nice theoretical interpretation. Here is the reference: Bao Wang, Binjie Yuan, Zuoqiang Shi, Stanley J. Osher. ResNets Ensemble via the Feynman-Kac Formalism to Improve Natural and Robust Accuracies, arXiv:1811.10745, NeurIPS, 2019

Overall, since this work is a straightforward integration of some existing work, I think this
paper lack novelty. Please address the above questions in rebuttal.

**Experience Assessment:**

I have read many papers in this area.

**Review Assessment: Checking Correctness Of Derivations And Theory:**

I assessed the sensibility of the derivations and theory.

**Review Assessment: Checking Correctness Of Experiments:**

I carefully checked the experiments.

**Review Assessment: Thoroughness In Paper Reading:**

I read the paper thoroughly.

---

> ### Author Response · Authors · 2019-11-15
> **Thank you for your review**
>
> We thank the reviewer for the insightful comments and questions. Please also see our general response above.
>
> Q1: Thanks for the suggestions. We will revise the abstract in later revision.
>
> Q2: Sorry for the confusion. M is T(x), which is a mixed random variable.
>
> Q3: Fig. 3 shows an example of CDF of a mixed random variable M to better understand T(x). Mixed random variables are neither discrete random variables nor continuous random variables (e.g., the sum of a geometric random variable and a Gaussian random variable).
>
> In Fig. 3, M=X \mathbb{I}(X !\in [0.95,2.95])+Pr(X \in [0.95, 0.95+2/3])*\delta(x;a=0.95+2/3) +Pr(X \in [0.95+2/3, 0.95+4/3])*\delta(x; a=0.95+4/3)+Pr(X \in [0.95+4/3, 0.95+2])*\delta(x; a=2.95). (\delta(x;a) is a dirac delta function, X ~ Exponential(1)). Similarly, T(x) is a mixed random variable, and follows a similar CDF.
>
> Minor measure-theoretic clarification: by definition, a mixed random variable does not admit a probability density function, although a mixed random variable can still have continuous range.
>
> Q4: Yes, the multi-class setting is developed in Theorem 1. Note that Pa and Pb in Theorem 1 denote the prediction probabilities for the most probable and the second most probable classes, respectively, in a multi-class setting.
>
> Q5: Wang et al. developed a theoretically motivated approach to improve ResNet models. However, such improvement cannot be practically certified: it relies on an attack algorithm (e.g., PGD) to show robustness. In contrast, we can compute a robustness certificate, which *proves* that no adversary exists within the certified region. We have updated our paper and made a clarification. Thank you for your reference!

---

### Official Review · AnonReviewer3 · 2019-10-23
**Official Blind Review #3**

**Rating:** 3

**Review:**

Summary.

The authors propose a new certified classifier in \ell_1 norm that is tight. That is to say, upon smoothing a given classifier f with Laplacian noise, a smoothed version of that classifier (probabilistic maximum majority vote) is certified with a radius measured in \ell_1 norm. The authors show that this bound is tight for binary classifiers. These results are complementary to Cohen et al. results.

Major comments.

1) The major contribution of this paper is the tightness under the \ell_1 norm for a binary classifier. I do not find this particularly significant. The question is of what value is such a result other than a mathematical exercise. For instance a good justification that the paper is lacking could be one where authors show that their radius is indeed tighter than all other works. The paper still lacks this (I will elaborate on this later), although, their bounds are indeed tighter than Lecure's et al. Since it is not clear whether or not the new certified smoothed classifier has indeed the largest radius among all other works, then at least a justification for why would one prefer a Laplacian noise of a Gaussian noise. Why is Gaussian smoothing sufficient for this purpose given that we do not know for sure that the radius is larger?  What value/advantages does this add? The authors motivate their work by saying deriving the tightest \ell_1 is difficult due to the "asymmetry"  of the norm. While I do agree on this; however, this is not enough motivation as we we are doing doing abstract maths here.

The new derived radius is not really comparable to the Gaussian radius with \ell_2 radius and this is my major concern. By norm equivalence, we have that \ell_2 \leq \ell_1 \leq \sqrt{n} \ell_2 where n is the dimension. That is to say that the radius computed with \ell_1 is larger than the \ell_2 in some cases by a square root of dimension. The authors can correct me on this if I'm wrong, but for a fair comparison in worst case sense the radius of Cohen et al. should be scaled by \sqrt{n}. In such a scenario, it is really difficult to understand when does it make sense to tackle such a smoothing technique as opposed to Gaussian smoothing.

I would not have asked the authors about such a question if the authors derived generic radius under \ell_p smoothing (which is difficult of course). To this end, I believe since the motivation is not clear nor the results are generic enough, I find the work incremental specifically after noting that the radius can be deduced from the work of Li et al. where the main contribution here is the tightness of the radius for a binary classifier.


Moreover, I believe the paper still requires some polishing in terms of writing and presentation.

Some more comments.

I believe the paper can benefit from some rewriting. Here is a list of things the authors can do to improve the paper.

1) Define what M is, page 3 "and it is easy to see that M is a mixed random variable". I believe the authors meant T(x).
2) The figures are hardly readable. For instance, authors can perhaps increase the legend's font size in figures 4. Also the chosen colors are suboptimal (perhaps the line width of the plots) should be increased.
3) The section below Theorem 3 should be moved up to before Theorem 3 as this discusses the proof of Theorem 2. Once a Theorem is presented, the proof sketch should follow.
4) Experiments on the undefended classifier has to be in Figures 6  7 and 8.
5) Lastly, why are comparison between Cohen et. al. and Lecuyer et. al. in Figure 6 inconsistent with Figure 5 of Cohen et al.

**Experience Assessment:**

I have published one or two papers in this area.

**Review Assessment: Checking Correctness Of Derivations And Theory:**

I assessed the sensibility of the derivations and theory.

**Review Assessment: Checking Correctness Of Experiments:**

I carefully checked the experiments.

**Review Assessment: Thoroughness In Paper Reading:**

I read the paper thoroughly.

---

> ### Author Response · Authors · 2019-11-15
> **Thank you for your review**
>
> We thank the reviewer for the insightful comments and questions. Please also see our general response above.
>
> Re “after noting that the radius can be deduced from the work of Li et al.”:
>
> This seems unfair for evaluating this work. After the first proof of tight results with Gaussian distribution by Cohen et al. (2019), Levine et al. (2019) and Salman et al. (2019) find simpler ways to derive the certificate by other approaches. These follow-up works do not invalidate Cohen et al. (2019). Similarly, Li et al. did not have the Laplace result before the dissemination of this work, and thus their deduction does not invalidate this work (let alone the fact that we even prove the tightness).
>
> Re “a justification for why would one prefer a Laplacian noise of a Gaussian noise”:
>
> One justification is that, since Laplace distributions puts more weight on the center than Gaussian distributions, Laplace noises are less prone to (negatively) impacting the prediction of the base classifier f than Gaussian noises. Indeed, taking a ResNet110 model on CIFAR-10 (trained without smoothing), we can obtain 24.8% accuracy by using a Laplace noise (variance = 0.12), while the Gaussian noise with the same variance would yield 23.7% accuracy. Here the accuracy is computed with respect to predictions of the base classifier instead of the labels (to illustrate how the smoothing impacts the predictions).
>
> We formalize the intuition in terms of the sensitivity of the noise distributions with respect to their hyperparameters (\lambda and \sigma), and prove that the Laplace noise is less sensitive than the Gaussian noise in terms of negatively impacting the base classifier f. This implies that it is easier to set the hyperparameter for the Laplace noises than Gaussian noises. For a detailed justification, please see Appendix D in the updated version.
> We do acknowledge that the two distributions are equally competent since the certifiable range are both [0, \infty). However, the resulting certificates of the two distributions are quite different in practice. An analogy would be architecture design in deep learning research. While existing architectures already exhibit universal approximation / turing completeness, new architectures with suitable inductive bias still improve the empirical performance quite a lot. Here Laplace noises can be regarded as an infinite mixture of L1 balls, which may be a suitable inductive bias for L1 robustness. Empirically, we indeed found that Laplace noises are much better than Gaussian noises for L1 robustness.
>
> Technical clarification: the L1 and L2 certificates of the Gaussian noise are exactly the same (there is no \sqrt{n} scaling). The reason is that given any L1/L2 radius r, we can show that the perturbation [r, 0, 0, …, 0] will be a theoretical worst case for Gaussian noises. As a result, the worst cases in L1 and L2 coincide, so the resulting certificates are exactly the same. One may prove the result by the fact that all the points within an L2 sphere have the same worst case prediction value under Gaussian noise (see Cohen et al.).
>
> Q1-Q4. Sorry for the confusion. M is T(x). We have corrected some figures and rearranged our writing. Thank you for the suggestions.
>
> Q5: Inconsistency between Cohen et. al. and Lecuyer et. al. in Figure 6 with Figure 5 of Cohen et al.
> A: Cohen et. al. shows the result of \ell_2 norm radius, while ours shows the \ell_1 norm radius. Lecuyer et al. have two certificates (for \ell_1 and \ell_2, respectively). In our paper we use the \ell_1 version while in Cohen et al. they use \ell_2 version, so they look different.

---

### Official Review · AnonReviewer2 · 2019-10-27
**Official Blind Review #2**

**Rating:** 6

**Review:**

The paper provides a random smoothing technique for L1 perturbation and proves the tightness results for binary classification case. Overall, there are some new results in this paper -- establishing a new certificate bounds for L1 perturbation model. However, I have several concerns about whether this contribution is significant enough:

Random smoothing has been studied extensively recently and the proof technique in this paper is not so different from previous papers (Cohen et al, Li et al). Also, there were L0 perturbation bounds proposed by (Leet et al). Therefore, although I agree that a tighter certified bound compared to (Lecuyer et al) is new, the paper seems to be a bit incremental. It will be more interesting to see if the proposed technique/theorem can be used for a wider range of norms.

Also, it may be more interesting to add some discussions about why L1 perturbation is important for image classification (is it more human-imperceptible?)

=======

I have checked the rebuttal and other reviewers' comments. Although there are interesting components in this paper, I do agree that the paper is incremental given that many random smoothing methods have been proposed recently for L2, L_infty norms. Therefore I think this is a borderline case and will be ok with rejection.

**Experience Assessment:**

I have published in this field for several years.

**Review Assessment: Checking Correctness Of Derivations And Theory:**

I assessed the sensibility of the derivations and theory.

**Review Assessment: Checking Correctness Of Experiments:**

I assessed the sensibility of the experiments.

**Review Assessment: Thoroughness In Paper Reading:**

I read the paper at least twice and used my best judgement in assessing the paper.

---

> ### Author Response · Authors · 2019-11-15
> **Thank you for your review**
>
> We thank the reviewer for the insightful comments and questions. Please also see our general response above.
>
> Q: Novelty and significance
> A: One part of our proof indeed uses the same Neyman-Pearson Lemma as (Cohen et al.), but our tightness proof result is new and cannot be derived from the existing approaches.
>
> Q: Importance of L1 perturbation
> A: Our explanation is that L1 distance is easier to interpret than L2 distance since L1 is simply the summation of absolute values without a nonlinear square root. Also, L1 has been widely studied in literature for measuring sparsity (thus connecting to sparse adversarial perturbations).

---

### Public Comment · ~Bai_Li1 · 2019-10-01
**a connection to existing work**

Thank you for the interesting work.

I would like to point out that in equation (1), while the first part λ/2*log(PA/PB) is equivalent to the bound from Lecuyer et al. (2019), the second part −λ log(1 − PA + PB) can be derived from Li et al. (2019) in Lemma 1 when alpha->∞, noticing the Renyi divergence of Laplacian distributions is 1/(α−1)log(α/(2α−1)exp((α−1)*R/λ)+(α−1)/(2α−1)exp(−α*R/λ) which converges to R/λ when alpha->∞. It also gives the same tight bound in the binary case where R = −λ log[2(1 − PA)].

It would be a great addition to your paper if you can make this connection clear. Thank you!

[1] Mathias Lecuyer, Vaggelis Atlidakis, Roxana Geambasu, Daniel J Hsu, and Suman Jana. Certified
robustness to adversarial examples with differential privacy. ieee symposium on security and
privacy, 2019.

[2] Bai Li, Changyou Chen, Wenlin Wang, and Lawrence Carin. Second-order adversarial attack and
certifiable robustness. arXiv: Learning, 2018.

---

> ### Author Response · Authors · 2019-10-03
> **responses to "a connection to existing work"**
>
> Dear Bai,
>
> Thank you for the comment!
>
> In our paper, we have made it clear that the first part in our upper bound theorem is the same as Lecuyer et al. (2019). It is very nice to know that the second part can be derived using your framework based on Renyi divergence. We will definitely acknowledge that it is possible to derive that bound under your framework in the paper through later revision. It is indeed a great addition to our paper to diversify the methods for deriving robustness certificates. Thank you!
>
> However, in order to make it clear (for the reviewers), we want to emphasize that this work is the first work to establish the certificate (Eq. (1)), no matter how it can be derived. Moreover, your previous paper does not subsume our results for the following reasons:
>
> 1. One of our main contributions is proving the *tightness* of the L1 certificates, including both upper and lower bounds. Similarly, Cohen et al. (2019) use the same algorithm as yours (i.e., Gaussian perturbation), but their results are still very interesting, because they were able to prove that their certificate is tight on L2. In our case, while the L1 bound may be derived in different ways (which is not established in the literature, though), we can further prove that the bound is tight, which is new and non-trivial.
>
> 2. In your paper, you were analyzing Gaussian perturbations on L2, but our paper uses Laplace distribution on L1. To use your framework to prove our upper bound results, one needs to rewrite the proof for your theorem 2 on Laplace distribution, and pick alpha->\infty for Lemma 1. In other words, although your proof framework is handy, our upper bound is not a trivial corollary of your theorem.
>
> Best Regards,
> Authors

---

> > ### Public Comment · ~Bai_Li1 · 2019-10-03
> > **responses**
> >
> > Thank you for the responses!
> >
> > I definitely agree that proving the tightness of the L1 certificates is an important contribution. I am not aware of such a tightness, although I have had the same bound. We are going to update our paper and acknowledge your results for the comprehensiveness.
> >
> > Best,
> > Bai

---

### Public Comment · ~Anthony_Wittmer1 · 2019-10-10
**A closely related paper**

Great work and I really enjoy reading it.

However, previous work has studied the robustness theory of randomization techniques on the general family of exponential distributions. Please check out this paper [1], where the randomized models with the Laplace distributions are also considered.

 In my opinion, a discussion/comparison seems due.

[1] Theoretical evidence for adversarial robustness through randomization. NeurIPS 2019

---

> ### Author Response · Authors · 2019-10-15
> **responses to "A closely related paper"**
>
> Dear Anthony,
>
> Thank you for the reference! The results in Pinot et al. (2019) are very great. The work addressed adversarial robustness for risk (expectation over data distribution). We will definitely add a discussion paragraph in the next revision.
>
> However, the paper (Pinot et al., 2019) is fundamentally different from ours. We work on robustness certificates for any (x_i, y_i) pairs, while Pinot et al. (2019) work on robustness guarantee for risk. Specifically, for every given x_i, we can really compute a radius R, such that for any perturbation \delta s.t. \|\delta\|_1 < R cannot alter the prediction (i.e., we guarantee g(x_i) = g(x_i + \delta)). Pinot et al. (2019) gave robustness guarantee for risk (including generalization gap), and they did not provide robustness certificates.
>
> As a side node, we would like to point out that we also proved the *tightness* of our L1 certificates, including both upper and lower bounds, which is new and non-trivial. Pinot et al. (2019) did not show tightness results.
>
> Best Regards,
> Authors

---

### Author Response · Authors · 2019-11-15
**General response**

We thank the reviewers for the insightful comments and questions.

We would like to clarify the novelty and significance of this paper.

This is an initial but important attempt towards tight results for general Lp norm and other distributions that inevitably involve mixed random variable analysis (cf. Gaussian and discrete) and asymmetric norms (cf. L2 and L0). While existing approaches fails in these challenging cases, our approach illustrates that tight results are still viable, and shines light on how these challenging cases can be tackled in general.

This work should be treated as an (Laplace, L1) analogy to the (Gaussian, L2) case proved by (Cohen et al.), which also improves (Lecuyer et al.) and proves that their result is tight.

---

### Decision · Program_Chairs · 2019-12-19

**Decision:**

Reject

**Comment:**

After reading the author's response, all the reviewers agree that this paper is an incremental work. The presentation need to be polished before publish.